# Cholera in the Time of MINUSTAH: Experiences of Community Members Affected by Cholera in Haiti

**DOI:** 10.3390/ijerph19094974

**Published:** 2022-04-20

**Authors:** Susan A. Bartels, Georgia Fraulin, Stéphanie Etienne, Sandra C. Wisner, Sabine Lee

**Affiliations:** 1Departments of Emergency Medicine and Public Health Sciences, Queen’s University, Kingston, ON K7L 4V7, Canada; 2Faculty of Arts and Science, Queen’s University, Kingston, ON K7L 3N9, Canada; 16gef1@queensu.ca; 3Komisyon Fanm Viktim pou Viktim (KOFAVIV), Port au Prince, Haiti; estephie2011@yahoo.fr; 4Institute for Justice & Democracy in Haiti, Marshfield, MA 02050, USA; sandra@ijdh.org; 5Department of History, University of Birmingham, Edgbaston, Birmingham B15 2TT, UK; s.lee@bham.ac.uk

**Keywords:** cholera, diarrhea, Haiti, MINUSTAH, peacekeeping, United Nations

## Abstract

In 2010, Haiti experienced one of the deadliest cholera outbreaks of the 21st century. United Nations (UN) peacekeepers are widely believed to have introduced cholera, and the UN has formally apologized to Haitians and accepted responsibility. The current analysis examines how Haitian community members experienced the epidemic and documents their attitudes around accountability. Using SenseMaker, Haitian research assistants collected micronarratives surrounding 10 UN bases in Haiti. Seventy-seven cholera-focused micronarratives were selected for a qualitative thematic analysis. The five following major themes were identified: (1) Cholera cases and deaths; (2) Accessing care and services; (3) Protests and riots against the UN; (4) Compensation; and (5) Anti-colonialism. Findings highlight fear, frustration, anger, and the devastating impact that cholera had on families and communities, which was sometimes compounded by an inability to access life-saving medical care. Most participants believed that the UN should compensate cholera victims through direct financial assistance but there was significant misinformation about the UN’s response. In conclusion, Haiti’s cholera victims and their families deserve transparent communication and appropriate remedies from the UN. To rebuild trust in the UN and foreign aid, adequate remedies must be provided in consultation with victims.

## 1. Introduction

### 1.1. Haiti Background

Having experienced decades of political instability, weak governance, foreign exploitation and repeated natural disasters [1], Haiti has been labelled the “poorest country in the western hemisphere” with a 2018 Human Development Index of 0.503—ranking 169 out of 189 countries and territories [2]. Haiti’s latest official estimates in 2012 suggested that over 6 million people lived below the poverty line of USD 2.41 per day, and over 2.5 million fell below the extreme poverty line of USD 1.12 per day [3]. Haiti has experienced high levels of civil and political unrest, organized crime, coups d’état, political leader assassinations, and rapid political turnovers [4], leading to the sanction of the United Nations Stabilization Mission in Haiti, known by its French acronym, MINUSTAH [5], which was operational between 2004 and 2017. As a peace-support operation (PSO), MINUSTAH’s initial purpose was to complement the actions of the state by protecting civilians from the threat of armed gangs [6].

A 7.3 magnitude earthquake struck Haiti on 12 January 2010 with the epicentre located near its densely populated capital, Port-au-Prince. The earthquake claimed approximately 300,000 lives, caused about USD 8 billion in damage, and left 1.3 million people living in temporary tented shelters in the greater Port-au-Prince area [7]. Haiti’s already weak health care and public health infrastructure were also devastated, with more than 50 hospitals or health centres either collapsed or rendered unusable due to earthquake’s damage [7].

### 1.2. Cholera in Haiti

In October 2010, nine months after the catastrophic earthquake, the government of Haiti declared a cholera epidemic. With the first cases reported in the Haitian Central Plateau, cholera spread quickly to affect all departments across the country and by 2018, over 800,000 cholera cases had been reported and up to an estimated 10,000 people had died [8]. However, these figures may actually be underestimates due to limited diagnostic testing in certain parts of Haiti [9]. Cholera had not been documented in Haiti previously [10], and the epidemic occurred at a particularly vulnerable time for an impoverished country in the midst of dealing with a major earthquake. Initially, there was significant debate about how *V. cholerae* was introduced into Haiti, with evidence pointing towards a Nepalese peacekeeping base having contaminated a tributary of the Artibonite River with its wastewater. The United Nations (UN) peacekeeping base was subsequently accepted as the source of cholera in Haiti when genomic sequencing confirmed an identical match between the Haitian and Nepalese cholera strains [11,12].

After years of denial and silence, in December 2016, the UN formally apologized for its role in the cholera epidemic in Haiti. In doing so, the UN declared a $400 million two-track approach to assistance in Haiti called “A New Approach to Cholera in Haiti” (New Approach) implemented through the United Nations Haiti Cholera Response Multi-Partner Trust Fund (MPTF) [13]. Track one included an ongoing effort to reduce the incidence of cholera by improving access to healthcare and addressing sanitation and water issues in Haiti, while track two included a promise of “material assistance and support to those Haitians most directly affected by cholera”. Calling on their “moral responsibility,” UN Secretary-General Ban Ki-moon encouraged voluntary funding from UN member states through the UN Haiti Cholera Response MPTF [14]. However, there were no measures put in place to ensure the adequate funding of the plan. In 2017, when the voluntary contributions had not been realized, the General Assembly of the UN asked member states to reallocate $40 million of unspent funds from MINUSTAH to address cholera in Haiti [15], with 31 countries having approved reallocation of the balance [16]. In June 2020, approximately $20.7 of the requested $400 million had been raised for the UN Haiti Cholera Response MPTF (5.2%) and the total use of these funds was only 50.4% [16].

Only five academic studies with empirical research on the perceptions of local Haitians about the cholera epidemic have been published. The data for two articles were collected in 2011 before the UN’s role in introducing the bacterium had been established [17,18]. The other three articles all focused on knowledge, attitudes and practices (KAP) related to water, sanitation and hygiene (WASH) in the midst of the cholera outbreak [19,20,21]. Thus, the existing body of research about the cholera outbreak lacks empirical documentation of the experiences and perceptions of Haitian community members themselves.

### 1.3. Purpose

Much has transpired since the earlier articles on local perceptions of the cholera outbreak were published—undeniable genetic proof that the cholera strain was from Nepal, a formal apology by the UN for its role, a $400 million UN response plan for assistance in Haiti, and a lack of progress towards realizing that plan. Considering all that has transpired, our aim in the current analysis was to understand how community members experienced the Haiti cholera epidemic between 2010 and 2017 with the goal of guiding future humanitarian responses and informing how to better meet the needs of affected Haitian community members.

## 2. Materials and Methods

### 2.1. Design and Setting

We conducted a cross-sectional study using SenseMaker as a narrative capture tool. SenseMaker uses open-ended story prompts which allowed our participants to share narratives on a variety of topics, including cholera. From the larger dataset, the current analysis focused exclusively on those micronarratives that were about the Haiti cholera outbreak (*n* = 77).

Ten UN bases across seven locations were purposively selected for this research based on years of operation, base size, troop-contributing countries staffing the base, geographic variation within Haiti as well as rural/urban designation.

### 2.2. Participants

A convenience sample of participants was approached in public spaces including market areas, public transportation stops/depots, and shops within a 30 km radius of each chosen base. This convenience sample included men, women and adolescents who were out in the community and accessible for participation. Individuals had to be at least 11 years old to participate. Anyone over the age of 11 who was in a public space in the areas surrounding a UN base could have been approached about the study. A diverse range of participant subgroups were targeted for inclusion to capture a variety of perspectives.

### 2.3. Data Collection

Cognitive Edge’s SenseMaker is a mixed methods research tool that extracts meaning from micro-narratives shared by participants on a topic of interest (in this case interactions between Haitian community members and MINUSTAH personnel). Three prompting questions were presented (shown in Appendix A) and participants were asked to share a micronarrative in response to the story prompt of their choosing. All micronarratives were audio recorded on a tablet. Participants then responded to a series of pre-defined questions which contextualized the recorded micronarrative (e.g., how often do the events in the story happen) and provided demographic data.

Researchers with collective expertise on humanitarian crises, public health and SenseMaker methodology wrote the SenseMaker survey in English. It was translated to Haitian Kreyòl, independently back-translated and then checked for accuracy, with discrepancies resolved by consensus. A pilot test of the survey with 54 participants in Haiti improved clarity, ease of response, participant comfort and translation inaccuracies.

We partnered with two local institutions, Komisyon Fanm Viktim pou Viktim (KOFAVIV) and the former Enstiti Travay Sosyal ak Syans Sosyal (ETS), to purposively select 12 research assistants (eight female and four male). The KOFAVIV research assistants were volunteers with the organization who had experience working with survivors of gender-based violence and the ETS research assistants were undergraduate social work students. Immediately before data collection, research assistants completed a four-day training session on SenseMaker methodology, informed consent, research ethics, a detailed question-by-question review of the survey, data uploading, as well as on the management of adverse events and program referrals. Research assistants used a standard script to explain the study design and intent to potential participants. Single, face-to-face interviews were conducted privately in Haitian Kreyòl and the shared stories were audio recorded on iPad Mini 4s. While the implementing study partner organizations may have been known to some participants, there was no prior established relationship between the authors and participants. Micronarratives were transcribed and translated from Haitian Kreyòl to English by native Kreyòl speakers.

### 2.4. Analysis

The survey asked research assistants to identify whether pre-determined topics were discussed in the shared micronarrative (e.g., cholera). Participants were also asked who the shared story was about. Using these two multiple choice questions, we selected narratives for analysis. As illustrated in Figure 1, 487 of the 2541 collected narratives were about or mentioned cholera and of those 189 were deemed to have sufficient detail for a qualitative analysis. Of the 189, we retained a total of 77 narratives that were documented in first person, shared by a participant about someone in their household/family, or about a friend. Chosen narratives were thematically analysed according to Braun and Clark [22]. Working independently in Microsoft Excel spreadsheets, two authors (SB and GF) reviewed the entire transcript closely and then conducted open coding of each narrative. Using Saldaña’s proposed methodology, each researcher initially coded the data line-by-line to identify diverse feelings, ideas, and experiences from each participant’s shared story [23]. These first level codes were generated directly from the text. Both researchers then reviewed the entire transcript again and agreed on first level codes, each representing a singular experience, idea, or feeling. In the second level of analysis, we organized the initial codes into five categories concerning the experience of cholera in Haiti. The categories were not mutually exclusive, and therefore stories could be placed into more than one. Researchers then selected pertinent narratives from each category to ensure that a diverse selection of cholera-related experiences were included.

The researchers engaged in critical dialogue around all aspects of story selection, coding, and analysis, and triangulation between researchers was key. Using constant comparison, each code or micronarrative was considered in relation to previous and subsequent data, and each micronarrative was considered as a whole. We kept an audit trail of all levels of coding as well as relevant memos.

### 2.5. Ethical Considerations

No identifying information was recorded and all interviews were conducted in private. Informed consent was reviewed in Haitian Kreyòl and indicated on the tablet. Since the study involved minimal risk, written consent was waived. The interviews were brief (approximately 15 min each) and participants did not travel to take part. As such, no compensation was offered. Participants as young as 11 years old were included because they were known to be affected by peacekeeper-perpetrated SEA and were considered to be mature minors. It would have been unethical to have excluded them from the study. Since involving parents would possibly have introduced bias and put adolescents at risk of parental conflict and/or abuse [24], parental consent was not sought. The Queen’s University Health Sciences and Affiliated Teaching Hospitals Research Ethics Board approved this study, including the waiver of parental consent (protocol # 6025181). Local community partners provided important guidance on cultural sensitivity and ethical considerations.

## 3. Results

The highest number of stories about cholera were shared from Hinche (20.8%) and Port-au-Prince communes (Cité Soleil with 16.9% and Tabarre with 9.1%). The majority of participants were male (64.9%), had some/completed secondary education (45.45%) and reported that their households had an average level of income (58.44%). Table 1 provides demographic details of the study participants.

Building on an earlier thematic analysis that examined local perceptions about how cholera was introduced to Haiti in October 2010, the current qualitative analysis explored how community members experienced the cholera epidemic under the following five major themes, each presented with illustrative quotes.

### 3.1. Cholera Cases and Deaths

Some participants had themselves been infected with cholera while others either had a family member, friend or neighbour who had experienced the disease. For example, a female participant in Cité Soleil became ill with cholera while she was pregnant.


*I had cholera, I was 8 months pregnant and was about to die from it. I had two children at home, they were also about to die from cholera, because I was severely ill…*
ID330 Female participant in Cité Soleil.

Some participants had numerous family members who had become infected with cholera. For example, the following participant’s family experienced multiple cholera deaths, devastating his neighbourhood.


*My neighbourhood was devastated by cholera… Many people died in their houses, they were all my family: wife, son, as well as daughter and her husband. All of these people are dead. I heard this thing [cholera] is MINUSTAH’s fault but I don’t know.*
ID799 Male participant in Léogâns.

There were also several narratives about children becoming orphaned after losing their parents to cholera. The following woman in Port-au-Prince (ID661)had adopted one such child after his mother and father died in the epidemic, saying “this child that you see here… He has neither a mother nor father. I adopted him as my son”.

A variety of emotions were expressed around the cholera epidemic. The most predominant feelings were suffering, frustration, and fear. For instance, several participants spoke of feeling frustrated by the epidemic and about frustration experienced in the community.


*I lived through the time of cholera, and it was dramatic. Every day I saw the cars going by, the ambulances going by… The frustration was boiling daily in the neighbourhoods.*
ID1284 Male participant in Saint Marc.

Others were fearful of becoming infected with/dying of cholera. For example, this participant shared a narrative about his friend who left the hospital’s cholera ward after everyone else in his hospital room died overnight.


*…the next morning, he came home with an IV in his arm. I told him he should’ve stayed in the hospital, but he replied, “No way I could stay there because out of four of us in the room, three died, all except me. So I decided to leave”.*
ID1959 Male participant in Hinche.

### 3.2. Accessing Care and Services

The narratives revealed perspectives on how local Haitian community members experienced health services during the cholera epidemic. Health care infrastructure and resources are often limited at baseline, and this was particularly true after the January 2010 earthquake. The widespread cholera epidemic further stretched those resources and, in some cases, it was very difficult to access care. Three themes are discussed below, including barriers to accessing cholera care, reflections on care received during the epidemic, and burial practices.

Several participants discussed obstacles when accessing medical care, including finding transportation to hospital because drivers were fearful of contracting cholera.


*Well this older person had cholera… When I realized he couldn’t walk I had to carry him on my back because I couldn’t find a ride nor a motorcycle to transport him. Everyone was concerned and avoided contact with people with cholera for fear of becoming a victim.*
ID1959 Male participant in Hinche.

A man in Saint Marc stated explicitly that a lot of people were infected with cholera and that some died because they were unable to access medical care. This is consistent with what is known about cholera—it can lead to dehydration and death within a few hours if left untreated.


*In our neighbourhood, a lot of people were infected with this illness, cholera. Some people even ended up dying from this disease because they could not get medical care, they ended up dying.*
ID1228 Male participant in Saint Marc.

Others described a lack of medical personnel to care for cholera patients. One man in Port-au-Prince reflected on the cholera epidemic and reported “there were not enough doctors in the country”. Another participant described there not being any staff to assist his aunt when he took her to hospital.


*It was an aunt that was a victim of cholera. Well, when we ran to the hospital with her, it wasn’t easy to find help. Because there were too many cases at the hospital so in order to get help quickly, that just didn’t happen.*
ID230 Male participant in Port-au-Prince.

Several participants acknowledged the role that MINUSTAH played in helping Haitian community members access care during the cholera epidemic. For instance, the following participant described how peacekeepers helped to transport patients to hospital.


*I had a child that had cholera. I went to the hospital with him… MINUSTAH were in the neighbourhood and I didn’t see them do anything bad. They would run and grab and take a sick person to the hospital. They are always watching over us.*
ID911 Female participant in Léogâns.

Among participants who had lost family members or friends to cholera, a few talked about burial practices. The following woman described being given a certificate (presumably a death certificate) after her uncle died in hospital. She stated that his body was zipped in (likely a body bag) and he was sent out, possibly referring to having limited access to the corpse, as is often the case given the extremely infectious nature of *V. cholerae*.


*My uncle was dead—he had passed 22 days after which he was out of excrement and he bled. They gave us a certificate in the hospital of Sainte Catherine… They zipped him and sent him out.*
ID578 Female participant in Port-au-Prince.

The following participant discussed how the government took responsibility for burying cholera victims. He also shared his awareness that from an infection control perspective, it was important to prevent bodily fluids escaping from the corpse.


*…when people died at the hospital, they dumped them in the field over there. This means that everyone was afraid of them, so that it’s the government who took care of their burial. And they say you have to hide all the holes in the human body, wherever water can enter. It’s a poison.*
ID1857 Male participant in Hinche.

### 3.3. Protests and Riots against MINUSTAH

Some participants discussed outrage against MINUSTAH once it became known that UN peacekeepers were responsible for introducing cholera in Haiti. A variety of reactions against MINUSTAH were expressed and participants spoke of accountability. Our analysis identified themes around calls for justice as well as protests and riots against MINUSTAH.

In a call for justice, one participant discussed pleas on the radio for those who had experienced cholera to join organized protests. The following woman, with six family members who had experienced cholera, was highly motivated to protest for justice.


*All I know is that I had six people in my family who were victims of it: one adult and five children. This morning I heard on the radio that everyone who was a victim of MINUSTAH is going to protest on the streets… They must protest for justice… I would love to follow and participate to do just like everyone else who was a victim to show how deep it hurt…*
ID391 Female participant in Port-au-Prince.

Other participants used stronger language but were more aspirational in tone, such as the following woman in Cap-Haïtien who indicated that she wanted to revolt against MINUSTAH so that they would leave Haiti.


*A lot of Haitians lost their family, their mothers, their fathers, etc., but if I could revolt against them I would, so they can leave the country.*
ID2518 Female participant in Cap-Haïtien.

The protests seemed to be centred in Cap-Haïtien, where according to the following participant, residents were upset and very angry.


*The population was upset and very angry, they throw bottles and stone at the agents. A lot of protest in the streets, barrels and tires burning, and people from everywhere, every part of the society… They understood what it means to have a relative catch cholera and die.*
ID1827 Male participant in Hinche.

Other narratives described riots. The following participant shared how two men were shot by peacekeepers during a riot against MINUSTAH.


*A group of young men in the area said that MINUSTAH should go. These guys dug a deep hole in the streets’ intersection, and a MINUSTAH tank fell into it. But without hesitation when the tank crashed into the hole, the MINUSTAH kneeled down and shot two young men, one of them named [ ], he did not die, but broke his feet, and the other one died on the scene.*
ID2334 Male participant in Cap Haitian.

### 3.4. Compensation for the Cholera Epidemic

Many participants mentioned compensation in one way or another in their narratives, which were collected approximately seven months after the UN outlined its two track New Approach, including the provision of material assistance to individuals and families most affected by cholera. It is important to note that at the time this data were collected, no study participants or their families had received any compensation from the UN.

Some participants, particularly those more confident that MINUSTAH was responsible for the cholera epidemic in Haiti, talked about the importance of compensation.


*…according to research, they [MINUSTAH] are responsible for the cholera outbreak that was a tort that caused us to lose some family members. A lot of people were lost to it…It’s a really rough, unforgiving disease. I think that if they compensate us, the nation, it would be good, and we would not be so sorry that they came to Haiti.*
ID1311 Male participant in Saint Marc.

Similarly, the following participant spoke of the fact that there had been no reparations despite many Haitians having lost their lives to cholera.


*Everyone knows the problems brought by the disease that the Nepalese forces brought here with them, where many Haitians died because of the disease that the Nepalese forces brought with them. In spite of that, we see that there was no reparation [given] to compensate these people… this negative aspect is something Haitians will always remember, they’ll never forget that negative aspect of MINUSTAH…*
ID1056 Male participant in Saint Marc.

Some participants discussed having documentation to prove their eligibility for compensation related to the cholera epidemic. A few individuals, like the man in the following narrative, indicated that his documents were ready, and he directly asked the UN to compensate cholera victims.


*I am still waiting for compensation for my wife who had cholera. I have a paper that is my “record” so that if there is compensation, I will be able to collect my benefit… I am asking that the United Nation finds a way to compensate us because right now there are multiple children that are orphans. They are left without a mother or father because of the cholera outbreak in Haiti.*
ID1725 Male participant in Port Salut.

In contrast, the following participant recounted how community members affected by cholera had been asked to make copies of their birth certificates in anticipation of being paid damages. He concluded by speaking of the hardship experienced by Haitians and how no reparation or compensation had been received to date.


*Yes, I caught cholera... They had sent everyone to make a copy of their birth certificates, especially those who had been hit by cholera in order to get indemnity. Nothing had been done. It is not only me, so many people experienced that hardship… We haven’t seen any reparation, any compensation.*
ID1782 Male participant in Hinche.

A few individuals had heard that compensation had already been paid by the UN following the cholera epidemic. Although the UN promised to hold consultations with those affected by cholera, these consultations were not mentioned in our 2017 interviews.

One participant talked about a radio program that had announced the UN’s intention to compensate cholera victims. He shared his belief that some funds had already arrived. He concluded by saying that he did not know what the president would do with the money although it was unclear if this was speculation about how funds would be distributed or, due to corruption, concern that the funds would not reach cholera victims.


*…I heard it on the radio, now it’s Radio (Telezene) on 3 Martinal. I heard several countries in the United Nations were planning to send money for people who were victims of cholera. After that, they said there was money they had already sent. Well, there’s money also they’re going to send. I don’t know what the president is going to do with that.*
ID1284 Male participant in Port-au-Prince.

There were also a small number of participants who stated that although compensation for cholera had been provided, they had not been able to access those funds as a result of not having the proper documentation: “I could not go because I did not have the card to go and get the money” (ID330).

### 3.5. Anti-Colonialism

Given the outrage felt by many Haitians over the introduction of cholera by MINUSTAH peacekeepers, and in the context of Haiti’s colonial history, as well as US occupation from 1915 to 1934, some narratives included sentiments of anti-colonialism and patriotism. Many individuals talked about how MINUSTAH made Haiti less safe, and the following participant stated that if Haiti had its own armed forces, things would have been different.


*Instead of getting better security, we became less safe in the country. I think that if it was the armed forces of the republic that were present, none of these things would not have occurred.*
ID2445 Male participant in Cap-Haïtien.

The disrespect perceived by many participants also included a lack of respect for humanity and dignity. In the following story, a male participant referred to MINUSTAH dumping human waste directly in front of community members and filming as well as sharing videos of Haitian children eating discarded food.


*…Sometimes they show outright disrespect. Sometimes when they’re dumping their toilets, they come right by you and do it… Our children are eating rubbish because of the bad situation they are in. MINUSTAH officers take out their phone or cameras to film these children and show how they live in abject poverty. Our government must get rid of MINUSTAH because the nation can take charge of its own responsibilities. We demand that MINUSTAH leave the country.*
ID1793 Male participant in Hinche.

In reference to the UN, a number of participants more explicitly talked about the invasion of Haiti. In this narrative, the participant described Haitians as being under occupation by MINUSTAH.


*They’re the ones who dropped poop in the waters—that’s why we have what they call cholera…in Haiti they came to provide us backup in terms of security… We can’t work together, we’re under occupation.*
ID1025 Male participant in Léogâns.

Similarly, the following individual references UN personnel wanting to take over the land but clearly states that there will be no Haiti without Haitians.


*They are the one who brought it [cholera] here…We are Haitian, we are under Haitian administration. We are not in the country of the white people. We cannot be under their administration. The Haitian government has to talk to MINUSTAH in order to know whether they will let them kill us all… Because there is no Haiti without Haitians. But if they end up killing us all, they will take over the land.*
ID1818 Male participant in Hinche.

A number of participants detailed how they believed the Haitian government was failing them in the midst of the cholera epidemic. For instance, this man in Cité Soleil expressed that accountability was lacking partially because the Haitian government lacked the political will to take a stance.


*After giving us the sickness the Haitian government just sat and watched, they never said anything but they know very well that the MINUSTAH gave it… The UN is waiting for the state get involved in the matter, they see that the state says nothing so they also deny everything too because if our own people are unable to say something, they do not have to raise their voice for us.*
ID511 Male participant in Cité Soleil.

## 4. Discussion

As we approach the twelfth-year anniversary of Haiti’s cholera epidemic, the voices of Haitians who experienced the outbreak have been largely overlooked. To address this, we analyzed 77 narratives to document local experiences with and perspectives about cholera. Our findings demonstrate the fear, devastation and frustration that resulted from becoming ill with cholera and/or from cholera-related cases and deaths in the family and community. Feelings of fear and frustration were sometimes compounded by barriers to accessing medical care. The results also demonstrate widespread anger toward MINUSTAH, manifested by protests and riots against the PSO. While most participants believed that the UN should compensate cholera victims, there was confusion around required documentation and misperceptions that compensation had been provided in some areas of the country but not others. Finally, MINUSTAH’s culpability in the cholera epidemic provoked anti-colonial sentiments as well as calls for an end to the UN’s ‘occupation’ of Haiti, while acknowledging the failings and shortcomings of the Haitian authorities to provide for, and protect the rights of, its citizens.

### 4.1. Cholera-Related Reactions to MINUSTAH

Multiple participants expressed fear and loss, similar to that described in 2011 by Grimaud and Legagneur [18]. However, in our analysis, anger and frustration towards MINUSTAH were more prominent than feelings of fear and loss. Participants were angry that Nepalese forces had brought the bacterium to Haiti, resulting in backlash against MINUSTAH personnel. While an earlier study in Haiti had reported a perception that cholera was a foreign disease deliberately spread for political reasons, those perceptions were more general and did not implicate MINUSTAH and the UN directly [18].

Many participants in the current study were also frustrated by the UN’s lack of response and its failure to deliver the promised compensation. Protests and riots against MINUSTAH arose most commonly in Cap-Haïtien, likely related to the death of a local adolescent just prior to the cholera outbreak, which had already sparked anti-UN protests [25]. Protests against MINUSTAH were also reported in Hinche where community members threw stones at Nepalese infantry [25], and in Mirebalais where protesters reportedly shouted, “We have no water to drink. We have no choice but to drink the water from the river! Like it or not, the UN must go” [1].

### 4.2. Legacy of Colonialism

Local perceptions framed the tension between Haitians and the UN as akin to a Haitian/foreigner dichotomy and the UN were seen as colonisers or occupiers. This highlights that Haiti’s colonial experience and history of foreign interventions is still present in the national narrative and feeds into the interpretation of current events. The anti-colonial sentiment speaks to this mistrust of foreigners in general, and foreigners associated with or perceived to be part of the Global North, such as the UN, in particular. This sentiment emerged irrespective of the nationalities and ethnicities of the implicated soldiers who were sometimes referred to by participants as ‘blan’, a term used in Haiti to mean foreigners, without referring to a specific race or skin color. Some have argued that MINUSTAH’s predominant makeup of Latin American peacekeepers have allowed them to be seen by the Haitian community as less hostile and more empathetic, due to their shared colonial histories [26]. Despite this, our results highlight a Haitian/foreigner dichotomy with the Nepalese apparently included in the foreigner category, likely contributing to local perceptions that the UN is part of an ongoing colonization of Haiti.

It is also important to remember that over the years, MINUSTAH was accused of sexual misconduct against local community members including sex with minors, human trafficking, and sexual exploitation [27,28]. In fact, the original study from which these data were derived was designed to examine local perceptions about interactions between MINUSTAH personnel and local women and girls. Separately published analyses from the same dataset focused on sexual exploitation and abuse (SEA) by MINUSTAH personnel [29,30,31,32] as well as peacekeeper-fathered children in Haiti [33,34,35,36,37]. Finally, it is important to note that over the years, several humanitarian and foreign aid debacles have been documented in Haiti, both pre-earthquake [38] and post-earthquake [1]. These have included corruption within the aid sector, foreign meddling in Haitian politics, and malicious interventions motivated by self-interest, which have usually been at the expense of Haitian citizens. From this perspective, it is not at all surprising that Haitian community members perceived them as an “armed forced occupation” and called for their departure from Haiti.

### 4.3. UN Response

There were frequent calls for compensation because of the UN’s negligent introduction of cholera to Haiti, which is to be expected given the UN’s promise in late 2016 to provide material compensation to victims and their families. The reality is that the UN has foreclosed compensation through its internal claims settlement process as well as through an independent claims commission and independent courts, despite recognition that compensation is pivotal for an effective remedy under international law [39]. It was clear from our data that myths and misunderstandings around compensation to cholera victims prevail, with some participants believing that they had missed out on compensation because they did not have the proper documentation. Indeed, figuring out who was affected by cholera and which deaths were cholera-related may be challenging 10 years on. However, a 2019 report provided recommendations for an appropriate response based on the needs, expectations and priorities expressed by the victims [40].

Although it took considerable time after the start of the Haitian cholera epidemic, the UN did update its peacekeeping medical manual to acknowledge the health risks peacekeepers pose to the host communities, in addition to outlining guidelines for educating about and vaccinating against cholera prior to troop deployment [41]. It remains to be seen how effectively these changes will be implemented and concerns have been raised about the UN’s ongoing unsafe sanitation practices across its global peacekeeping missions [39]. Furthermore, because these updates were quite cholera-specific, in the midst of the current COVID-19 global pandemic [42], there are ongoing concerns that deploying UN peacekeepers may carry COVID-19 and introduce the virus into host communities, as was the case in South Sudan where the first three confirmed cases of COVID-19 were among foreign UN personnel [43].

### 4.4. Strengths and Limitations

There are a number of important study limitations. The findings are generated from a convenience sample of 77 participants and we did not collect data about non-responders. Therefore, the results are not representative and cannot be generalized. Additionally, SenseMaker narratives tend to be shorter and less detailed than those derived from more traditional qualitative research. As a result, our findings may not have fully captured all the nuances of cholera experiences among Haitian community members and data saturation may not have been reached. Because the narrative prompts were open, some participants talked about sexual misconduct, other wrongdoings, or indeed about the positive aspects of MINUSTAH. Furthermore, the data were collected in Haitian Kreyol and subtleties may have been lost in the process of translation. Travel restrictions related to insecurity and COVID19 have postponed the sharing of results directly with local community members. Finally, we recognise our positionality and are cognizant that since some team members are non-Haitian academics, the results are partially interpreted with our own inherent biases. However, Haitian team members help to mitigate this risk.

The research also has several noteworthy strengths. First, open story prompts allowed micronarratives about cholera to emerge more naturally from the overall lived experiences of participants. Since no participants were asked about cholera during the interview, the shared perspectives emerged naturally, highlighting that cholera continues to be an important issue in the minds of local Haitians. Second, diverse participant subgroups were included within the Haitian Central Plateau and elsewhere in the country to include a variety of experiences and perceptions. Third, to the best of our knowledge, there has been only a single publication examining the experiences of Haitian community members with cholera since the UN accepted responsibility in late 2016 and very few overall publications on Haitian perspectives regarding cholera. Given that cholera has lingered in Haiti and the UN’s slowly evolving response, our empirical data serves an important role in bringing forth the voices of affected Haitian community members to identify their ongoing needs and concerns.

## 5. Conclusions

To eradicate cholera in Haiti, there must be renewed commitment to working with Haitian community members and authorities to improve access to clean water and adequate sanitation. Furthermore, cholera victims and their families should be kept informed with clear and transparent communication from the UN. They should know where they stand, why compensation has not been delivered in accordance with their legal rights [39], and what they might expect in the future. To not provide this basic level of communication to the Haitian population is disrespectful and fuels animosity. The UN must follow through on its promises and fulfil victims’ right to an effective remedy and, as called for by the Institute for Justice and Democracy in Haiti (IJDH) and Bureau des Avocats Internationaux (BAI), it needs to reenvisage its response to victims and successfully model respect for the rule of law and acceptance of accountability. Finally, to re-build trust in the UN and foreign aid more broadly, victims need to be consulted, and future interventions must be sensitive to Haiti’s history of colonialism as well as foreign occupation and intervention.

## Figures and Tables

**Figure 1 ijerph-19-04974-f001:**
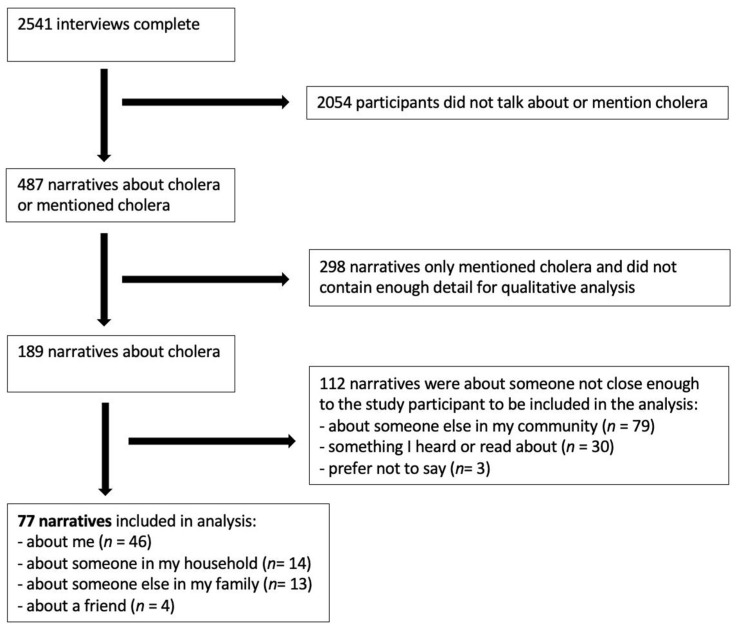
Narrative sampling process.

**Table 1 ijerph-19-04974-t001:** Participant Demographic Characteristics (*n* = 77).

Demographic		*n* (%)
Sex	Female	27 (35.1)
	Male	50 (64.9)
Age	11–17	7 (9.1)
	18–24	12 (15.6)
	25–34	23 (29.9)
	35–44	18 (23.4)
	45–54	8 (10.4)
	≥55	7 (9.1)
	Prefer not to say	2 (2.6)
Location	Hinche Cité Soleil Saint Marc Léogâns Cap-Haïtien Charlie Log Base/Tabarre Port Salut Morne Casse/Fort Liberté	16 (20.8) 13 (16.9) 12 (15.6) 12 (15.6) 10 (13.0) 7 (9.1) 6 (7.8) 1 (1.3)
Education	Some primary school	14 (18.2)
	Completed primary school	8 (10.4)
	Some secondary school	22 (28.6)
	Completed secondary school	13 (16.9)
	Some post-secondary school	9 (11.7)
	Completed post-secondary school	4 (5.2)
	No formal education	7 (9.1)
* Income Level	Poor	29 (37.7)
	Average	45 (58.4)
	Well-off	3 (3.9)

* Household income was assessed through a proxy measure which asked if the family owned any the following items: (1) mobile phone, (2) radio, (3) refrigerator/freezer, (4) any type of motorized vehicle, or (5) electricity/solar panels. Participants could choose as many items as was applicable or they could choose ‘none of the above’. Income was then categorized as follows: access to none or 1 of the 5 items was rated as ‘poor’, access to 2 or 3 of the items was rated as ‘average’ and access to 4 or 5 of the items was rated as ‘well-off’.

## Data Availability

Data are available on Figshare: 10.6084/m9.figshare.12388775.

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
