# Peer review of "Cholera in the Time of MINUSTAH: Experiences of Community Members Affected by Cholera in Haiti"

_ijerph, 2022, doi:10.3390/ijerph19094974_

Round 1

Reviewer 1 Report

Clearly this paper is interested and present merit for a final publication. But need some additional changes:

  1. In the abstract, please write in narrative style. Delete Background, results, please insert Qualitative methods are used... and  The conclusions suggest...
  2. In the structure of the paper. Please insert a strong introduction with the global significance of the research theme. A new section of theoretical background of the research from the medical geography... Renumbered the sections.
  3. In the methods, please discuss the qualitative or geo-ethnographic methodologies. Use references as Crang (2007), Hay, (2003), Hoggart et al (2002) and Pryke, Rose and Whatmore (2003) ...
  4. In the discussion, please acknowledge the limitations of the results and the validity for global scenarios.
  5. With this changes the paper wins in its scientific impact  

Author Response

Reviewer 1:

Clearly this paper is interested and present merit for a final publication. But need some additional changes:

1. In the abstract, please write in narrative style. Delete Background, results, please insert Qualitative methods are used... and  The conclusions suggest...

The structured headings have been removed on page 1 as per the reviewer’s recommendation.

2. In the structure of the paper. Please insert a strong introduction with the global significance of the research theme. A new section of theoretical background of the research from the medical geography... Renumbered the sections. 

This research was undertaken with a public health lens. It is not a medical geography paper nor are any of the coauthors from a geography field. Therefore, we have not included a medical geography theoretical framing.

3. In the methods, please discuss the qualitative or geo-ethnographic methodologies. Use references as Crang (2007), Hay, (2003), Hoggart et al (2002) and Pryke, Rose and Whatmore (2003) ...

We thank the reviewer for this suggestion. However, our research was conducted from a public health perspective rather than with a geography lens. Our qualitative methodology is referred to in the Methods on page 4, - we used an adaptation of thematic analysis as described by Braun and Clark (2006). Given that we did not conduct an ethnography or use methodologies described from the field of geography, it would not be appropriate to refer to the works of Crang, Hay, Hoggart, or Pryke, Rose and Whatmore.

4. In the discussion, please acknowledge the limitations of the results and the validity for global scenarios. 

The limitations for the study, including that the results cannot be generalized, are described in a dedicated paragraph in the Discussion on the bottom of page 12 and top of page 13.

5. With this changes the paper wins in its scientific impact  

We are grateful to the reviewer for these comments and suggestions to improve the manuscript.

Reviewer 2 Report

Cholera in the Time of MINUSTAH: Experiences of Community Members Affected by Cholera in Haiti

This is a very interesting manuscript, very complete and well written, which deserves to be published at the IJERPH. I have only some comments to improve its presentation.

I consider that the structure is clear and that each interview contributes to support each of the studied terms. Derived from the cholera outbreak in Haiti, the authors studied  the problems of the Haitians with the UN troops, with the UN itself, feelings of anti-colonialism and a certain feeling of anger at the lack of medical services and the lack of compensation (in money or services) from the UN, after causing Haitians the problem of cholera, caused by an irresponsible treatment of sewage in the Nepalese headquarters of the UN troops. The authors provide a clear Discussion and a very interesting Conclusion. After reading this excellent manuscript, I decided to contribute to solve only some formatting problems.

General comments: Some paragraphs are written in red ink. Perhaps it is a failure in the electronic system of the journal (?).

Lines 111-116. 2.2 Participants. “A convenience sample of participants…” I think it is necessary to be more explicit.

References. Please check that all journal names are abbreviated (e. g., references 8, 24, 31), including the use of periods (e. g., references 17, 19, 20).

Author Response

Reviewer 2:

1. This is a very interesting manuscript, very complete and well written, which deserves to be published at the IJERPH. I have only some comments to improve its presentation.

I consider that the structure is clear and that each interview contributes to support each of the studied terms. Derived from the cholera outbreak in Haiti, the authors studied the problems of the Haitians with the UN troops, with the UN itself, feelings of anti-colonialism and a certain feeling of anger at the lack of medical services and the lack of compensation (in money or services) from the UN, after causing Haitians the problem of cholera, caused by an irresponsible treatment of sewage in the Nepalese headquarters of the UN troops. The authors provide a clear Discussion and a very interesting Conclusion. After reading this excellent manuscript, I decided to contribute to solve only some formatting problems.

We appreciate the reviewer’s recommendations – thank you.

2. General commentsSome paragraphs are written in red ink. Perhaps it is a failure in the electronic system of the journal (?).

Thank you for raising this. Perhaps it was the earlier round 1 revisions. There does not appear to be any text in red in the current version.

3. Lines 111-116. 2.2 Participants. “A convenience sample of participants…” I think it is necessary to be more explicit.

An additional statement has been added to the Methods on page 3 to make the recruitment more explicit: “This convenience sample included men, women and adolescents who were out in the community and able to be approached for participation.”

4. References. Please check that all journal names are abbreviated (e. g., references 8, 24, 31), including the use of periods (e. g., references 17, 19, 20).

The journal names have been abbreviated as suggested.

Editors Comments:

The manuscript has improved and most of the reviewers' questions and concerns have been addressed. The only part that has not been reduced as requested is that of the results. However, I think it is suitable to be sent for peer-reviewers.

The Results had been trimmed somewhat in the first round of revisions and we have further trimmed them again. Please see pages 6 – 10.

Round 2

Reviewer 1 Report

All recimendations suggest in the firt round

Author Response

Just some points I would like to raise, asking the Authors to pay attention to reviewing them:

- Line 112  “A convenience sample of participants was approached in naturalistic location (including market areas …..”. I think that it could be useful to explain what  the convenient approach in recruitment the people was. I wonder if “naturalistic locations” is the appropriate term. 

We have removed the term ‘naturalistic locations’ and updated the text on page 3 to read:

A convenience sample of participants was approached in public spaces including market areas, public transportation stops/depots, and shops within a 30 km radius of each chosen base. This convenience sample included men, women and adolescents who were out in the community and able to be approached for participation. Individuals had to be at least 11 years old to participate. Anyone over the age of 11 who was in a public space in the areas surrounding a UN base could have been approached about the study. A diverse range of participant subgroups were targeted for inclusion to capture a variety of perspectives. 

- line 193 Table 1 “Income level”; what was the reference for the classification in “poor”, “average”, “well-off”? 

- Thank you for pointing out this omission. We have now added a footnote under Table 1 on page 6 which reads:
*Household income was assessed through a proxy measure which asked if the family owned any the following items: 1) mobile phone, 2) radio, 3) refrigerator/freezer, 4) any type of motorized vehicle, or 5) electricity/solar panels. Participants could choose as many items as was applicable or they could choose ‘none of the above.’ Income was then categorized as follows: access to none or 1 of the 5 items was rated as ‘poor,’ access to 2 or 3 of the items was rated as ‘average’ and access to 4 or 5 of the items was rated as ‘well-off.’

- line 452 Reference 33. In the reference list, the article by Schwartz TT 2008 is reported as 33, which does not seem to correspond to what is written in the text (“2011 by Grimaud and Legagneur”). 

We appreciate that you caught this. In the various rounds of revisions there was a mix-up in the references. This has been fixed and other citations have also been confirmed. 

- Line 549 Reference 41 is not reported in the reference list.  

This has been fixed and other citations have also been confirmed.

- Line 552 IJDH and BAI. Acronyms should be made explicit. 

Both acronyms have not been written out in full on page 13.

- References No 30 and 40 are missing both in the text and in the list of references. 

This has been fixed and other citations have also been confirmed.

This manuscript is a resubmission of an earlier submission. The following is a list of the peer review reports and author responses from that submission.

Round 1

Reviewer 1 Report

Need to improve on the overall presentation and methodology.

Author Response

Thank you for reviewing our manuscript. The Introduction, Results and Discussion have all been
shortened and streamlined to improve readability.

Reviewer 2 Report

First of all, the article is too long.

Second one is that in the result section the authors include the statement of representative participants. If they summarize the result on the basis of group and  location on their own it would be better.

Discussion needs to be concise.

Author Response

First of all, the article is too long.
- The Introduction, Results and Discussion have all been shortened and streamlined to improve
readability. The revised article is now just over 16 pages (down from 21).
Second one is that in the result section the authors include the statement of representative participants.
If they summarize the result on the basis of group and location on their own it would be better.
- We are unclear about what is being asked. The results begin with two sentences that provide an
overview about the participants’ demographics with additional details included in Table 1. If the
suggestion was to present Table 1’s demographics by location, that would make the table exceptionally
large given that there 8 locations.
- The reviewer could also be referring to the demographic summary statement that follows each quote
(for instance, ID330 Female participant in Cité Soleil). However, this already includes the location. We
are not sure what ‘group’ refers to.
- If the editor believes that something needs to be adjusted in the Results, please provide further guidance
on what is being asked
Discussion needs to be concise.
- The Discussion has been shortened and streamlined to improve readability. The number of subheadings
has been reduced and some elements that were not as closely linked to the results have been removed
(ex. Water and sanitation).

Reviewer 3 Report

The article is too long (21 pages). The chapter of results in particular is too detailed since that it reports many original sentences  for every topic. This can be justified if there were many responders, but the sample counts only 77 responders. Please summarise the main contribution for every topic

Author Response

The article is too long (21 pages). The chapter of results in particular is too detailed since that it reports
many original sentences for every topic. This can be justified if there were many responders, but the
sample counts only 77 responders. Please summarise the main contribution for every topic.
- The Introduction, Results and Discussion have all been shortened and streamlined to improve
readability. The revised article is now just over 16 pages (down from 21) and some of the extraneous
details have been removed so that the main points are brought to the forefront.